# Effects of long-term and high-dose administration of glucocorticoids on the cranial cruciate ligament in healthy beagle dogs

**Masakazu Shimada**[1]*, **Koki Wada**[1], **Sachiyo Tanaka**[1], **Sawako Murakami**[1], **Nobuo Kanno**[1], **Kei Hayashi**[2], **Yasushi Hara**[1]

**1** Division of Veterinary Surgery, Department of Veterinary Science, Faculty of Veterinary Medicine, Nippon Veterinary and Life Science University, Musashino-shi, Tokyo, Japan, **2** Department of Clinical Sciences, College of Veterinary Medicine, Cornell University, Ithaca, New York, United States of America

\* masakazu42710@gmail.com

## Abstract

This study aimed to determine the effects of long-term and high-dose administration of glucocorticoids (GCs) on the histological and mechanical properties of the cranial cruciate ligament (CrCL) in healthy beagle dogs. A synthetic corticosteroid at 2 mg/kg every 12 h was administered for 84 days in nine dogs (18 CrCLs) (GC group). Twenty CrCLs from 12 healthy male beagles were used as the normal control (control group). CrCLs were histologically examined (n = 12 in the GC group and n = 14 in the control group) using hematoxylin-eosin, Alcian-Blue, Elastica-Eosin stains, and immunohistological staining of type 1 collagen and elastin. An additional 12 CrCLs were mechanically tested (n = 6 in the GC and n = 6 in the control groups) to determine failure pattern, maximum tensile strength, maximum stress, elastic modulus, and stress and strain at the transition point. The histological examination revealed a significant increase in interfascicular area and fibrillar disorientation at the tibial attachment in both groups. The ratios of mucopolysaccharide-positive area and positive areas of elastic fibers were significantly higher in the control group than in the GC group. The biomechanical examination demonstrated significantly lower stress at the transition point in the GC group than in the control group. The present study results indicate that high-dose corticosteroids may affect metabolism, such as mucopolysaccharides and elastic fibers production, although the effect on type 1 collagen production is small. These changes of the extracellular matrix had a small effect on the strength of the ligament. This study suggested that the ligamentous changes associated with GC are different from the degeneration observed in spontaneous canine CrCL disease.

## Introduction

Cranial cruciate ligament rupture (CrCLR) is a common cause of hindlimb lameness in dogs [1, 2]. Canine CrCLR is characterized by progressive chronic ligament degeneration with

**Data Availability Statement:** All relevant data are within the manuscript and its Supporting information files.

**Funding:** The author (Y.H) was funded by a Grant-in-Aid for Scientific Research from the Japan Society for the Promotion of Science (Grant number: 17K08114, URL:https://www.jsps.go.jp/english/e-grants/). The creation of the model dogs in this study was supported by the Grant. The funders had no role in study design, data collection and analysis, decision to publish, or preparation of the manuscript.

**Competing interests:** The authors have declared that no competing interests exist.

chondrometaplasia, and it is referred to as "cranial cruciate ligament disease" (CCLD) [3–6]. The ligament is composed of ligamentous cells and extracellular matrix (ECM), of which collagen comprises approximately 75% of the dry weight [7]. Approximately 85% of ligamentous collagens are type 1 collagen (COL1), which attributes to the ligament's resistance to tensile forces [7]. Cranial cruciate ligament (CrCL) degeneration with chondrometaplasia reportedly occurs with increased cartilage matrix, such as type 2 collagen and proteoglycans [4–6]. The aforementioned degeneration induces the deterioration of mechanical properties and progresses to CrCLR [5]. Despite the unclear pathogenesis of CCLD, researchers have identified numerous associated factors, such as genetics, age, body weight, endocrinological factors, neurological factors, and bone conformation [5, 8–11].

The physiological effects of GCs include metabolism, water and electrolyte balance, and immunomodulation [12, 13]. The effects from supraphysiologic doses of glucocorticoids include anti-inflammatory, immunosuppressive, and protein catabolic effects [12, 13]. There have been reports on suspected non-traumatic Achilles tendon rupture cases in humans, associated with long-term steroid therapy and Cushing's syndrome [14, 15] and an increased risk of patellar tendon rupture, associated with the intra-articular administration of steroids to the knee joint [16]. The endocrinology textbook said chronic hypercortisolism could exaggerate common problems, such as CrCLR and patellar luxation lameness [17]. To date, no study has yet examined the cause and effect relationship between CrCLR and GC in dogs.

The purpose of this study was to examine the CrCL of healthy beagle dogs that were administered a high dose of the GC. We hypothesized that GCs affect ECM production in CrCLs, thus resulting in ligament fragility. The effects of GCs on the CrCL were investigated by histologically evaluating the degree of CrCL degeneration and by analyzing the tensile biomechanical properties of the CrCL using a universal material testing machine.

## Materials & methods

### Animals

This study was approved by the Institutional Bioethics Committee of the Experimental Animal Committee (approval number: 2019S-60 and 2020S-25). Table 1 shows the individual information of the 21 male beagles involved in the study. Nine male beagles were orally administered a high dose of a synthetic corticosteroid (prednisolone) at 2 mg/kg every 12 h for 84 days (GC group). The model used for this study is that for cardiovascular research. A previous report examined a model in which dogs were administered a high dose synthetic corticosteroid at 2 mg/kg every 12 h for 28 days, which showed changes in echocardiographic cardiac

**Table 1. Individual information.**

| | Number of dogs | BW (kg) | Age (months) |
|---|---|---|---|
| **Individual information** | | | |
| Histological analysis | | | |
| GC group | 6 (12 stifles) | 10.8 (9.5–11.2) | 16(15–17) |
| Control group | 7 (14 stifles) | 11.9 (10.0–14.0) | 16 (16–24) |
| Biomechanical analysis | | | |
| GC group | 3 (6 stifles) | 11.0 (10.0–12.1) | 15 (15–15) |
| Control group | 5 (6 stifles) | 9.0 (7.5–10.2) | 14.5 (10–17) |

BW: body weight; GC: glucocorticoid.

Data are presented as median (minimum-maximum).

morphology and function; however, histological changes were not observed [18]. Past studies have suggested that chronic hypercortisolism can affect cardiac function and morphology [19, 20]. Our research group hypothesized that histological changes would appear with longer-term observation and conducted the study for 84 days (approval number: 2019S-72, 29S-37). In the GC group, six dogs (12 stifles) were used for histological analysis and three dogs (six stifles) for biomechanical analysis, respectively. Twenty stifles from 12 healthy male beagles euthanized for other unrelated approved studies (approval number: 2019J-29, 30K-9) were used as a normal control group (control group). Of these beagles, fourteen stifles from 7 dogs were used for histological analysis and 6 stifles from 5 dogs were used for biomechanical analysis, respectively. Beagles in both the GC and control groups were housed in cages (0.7 m wide × 1.2 m deep × 1.5 m high), and loads such as exercising on a treadmill were not provided. The dogs were kept in the cage for the whole experiment duration but could roam freely indoors for ~1 hour twice daily (morning and evening). All dogs were euthanized with an overdose of pentobarbital to minimize animal suffering.

## Histological and immunohistological analysis

Following euthanasia, the CrCLs were collected and fixed in 4% paraformaldehyde for 24 h. The paraformaldehyde was not changed during the process. Subsequently, a paraffin-embedded procedure was performed and thin longitudinal sections of each specimen were prepared. These specimens were used for hematoxylin-eosin, Alcian-Blue (AB), Elastica-Eosin (EE), and immunohistological staining.

Immunohistological staining was performed for COL1 and elastin. The sections were immersed in methanol containing 3% $H_2O_2$ for 30 min to avoid endogenous peroxidase activity. Antigen retrieval was performed by incubation in a citrate buffer (0.01 M, pH 6.0) for 60 min at 65˚C. While the COL1 sections were blocked with normal rabbit serum, those for elastin were blocked with normal goat serum for 30 min at room temperature before applying the antibodies. The sections were incubated overnight with primary antibodies against COL1 (1:500 dilution; ARG21965 Goat Polyclonal antibody recognized COL1, Arigo Biolaboratories Corp., Hsinchu, Taiwan) and elastin (1:100 dilution; ab21610 Rabbit Polyclonal antibody recognized elastin, Abcam, Cambridge, UK). To confirm the cross-reactivity, appropriate positive controls were included in each immunostaining protocol. The normal canine mammary gland was selected as the positive control for COL1. In contrast, the normal canine aorta was selected as the positive control for elastin (Fig 1). The sections were then incubated with a secondary

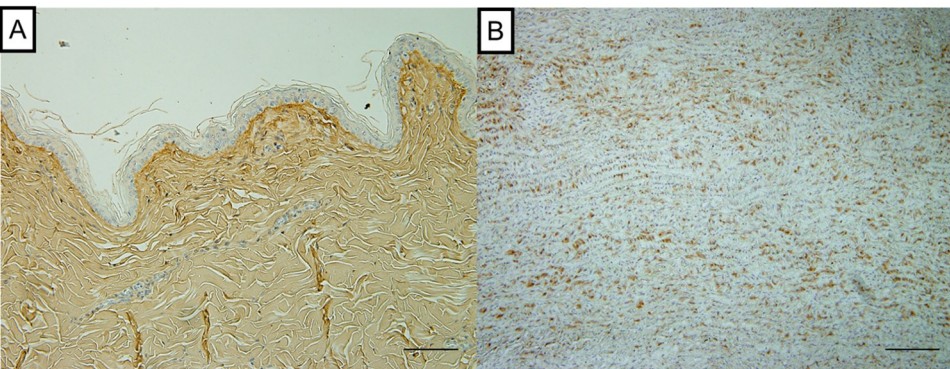

**Fig 1. Positive control of immunohistochemical staining.** A: A positive area is found in the dermis using the mammary gland as a positive control for type 1 collagen. B: A positive area is found in the intracellular area of the mesothelial area using the normal canine aorta, selected as the positive control for elastin.

antibody for 60 min at room temperature and subsequently stained with 3,3′-diaminobenzidine for 2 min. The slides were counterstained with hematoxylin.

The sections were fractionated into three compartments at the tibial, middle, and femoral sides. Three fields of 400x were randomly selected in each compartment and the cell density in the hematoxylin-eosin stained slides was assessed. AB stained slides were used to evaluate the mucopolysaccharide-positive areas and EE was used to evaluate the elastic fiber-positive areas stained with resorcin-fuchsin. For the immunohistological analysis, COL1 positive areas and the elastin positive cells were evaluated. The elastin positive cell density was calculated by dividing the total number of cells in the observed field of view by the area (cells/mm$^2$). The positive area ratio was calculated by dividing the positive area by the total area x 100. The positive cell ratio was calculated by dividing the number of positive cells by the total number of cells x 100. All data were analyzed using an image analysis software (ImageJ) by a single-blinded observer.

## Biomechanical analysis

**Specimen preparation.** Following euthanasia, the tibiofemoral joint was harvested along with the soft tissues surrounding the stifle joint. The specimens were wrapped in lactated Ringer's solution-soaked gauze and stored at -20˚C until the test day. All specimens were used within 1 year of the collection. After thawing at 4˚C a day before the test, all soft tissues except the CrCL were removed to create a bone-ligament-bone model. The femur and tibia of each stifle were then sectioned to a length of 10 cm from the joint line. Subsequently, the ends of the femur and tibia were potted in cylindrical molds of acrylic resin (GCOSTRON II I; GC Corporation, Tokyo, Japan). Lactated Ringer's solution was sprayed during the testing to prevent drying after thawing.

**Testing protocol.** A universal material testing machine (Autograph AGS-X, SHIMADZU Corporation, Kyoto, Japan) was used. The joint was set to 135˚, the angle in the standing position, and the CrCL was placed on the machine, the same direction as the proximal-distal direction of distraction. A preliminary load of 5 N was applied to release the creep, followed by a load of 10 N/sec [21]. An approximate straight line of the toe region was created in the range of 0.05 MPa to 0.2 MPa, which showed linear behavior in all specimens following preliminary loading. The transition point was determined, which was the intersection of the approximate straight line of the toe-region and the approximate straight line of the linear region (Fig 2) [22]. The following parameters were recorded and analyzed: the failure pattern, maximum tensile strength (N), maximum stress (MPa) [maximum tensile strength (N) / sectional area (mm$^2$)], elastic modulus [stress (N) / strain (%)], stress at the transition point (MPa), and strain at the transition point (%). The CrCLs were transected perpendicular to the longitudinal axis of the ligament with a scalpel at its attachments to femur and tibia, and the section was stained with India ink to measure the femoral and tibial cross-sectional areas using Image J (Fig 3). The average of the two cross-sectional areas from both sides was used as the cross-sectional area of the CrCL.

## Statistics analyses

SPSS software version 26 (SPSS Inc., Chicago, IL, USA) was used for all statistical analyses. Statistical analysis included comparison of the differences in the areas of each group from the histological and immunohistological analyses. The Kruskal Wallis test was followed by the Mann-Whitney test as a post-hoc test. In addition, the Mann-Whitney test was performed to compare the differences among the histological, immunohistological, and biomechanical analyses within each group. Differences were considered statistically significant at a P-value <0.05.

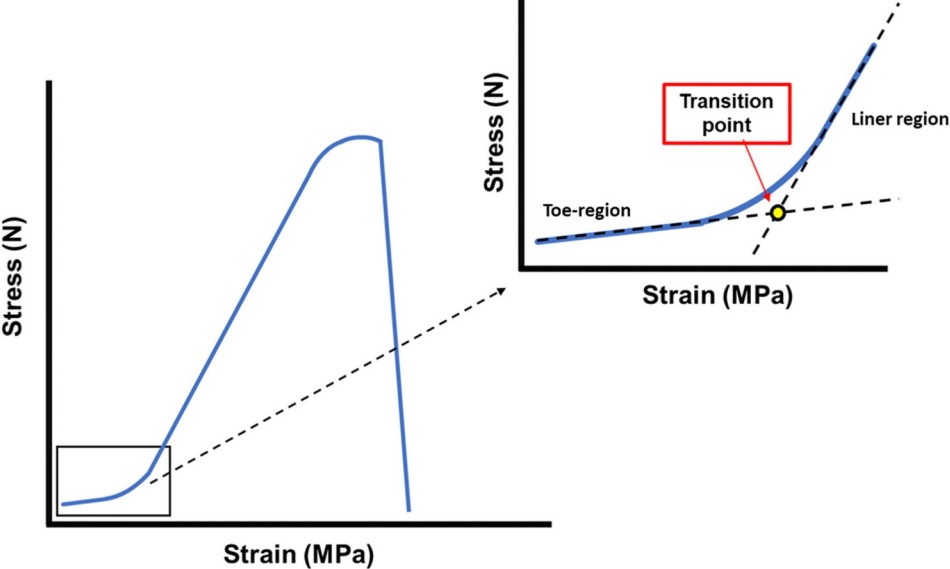

**Fig 2. The stress-strain curve.** The transition point (yellow dot) has been set at the intersection of the toe-region approximation line and the liner-region approximation line following the creation of the stress-displacement curve.

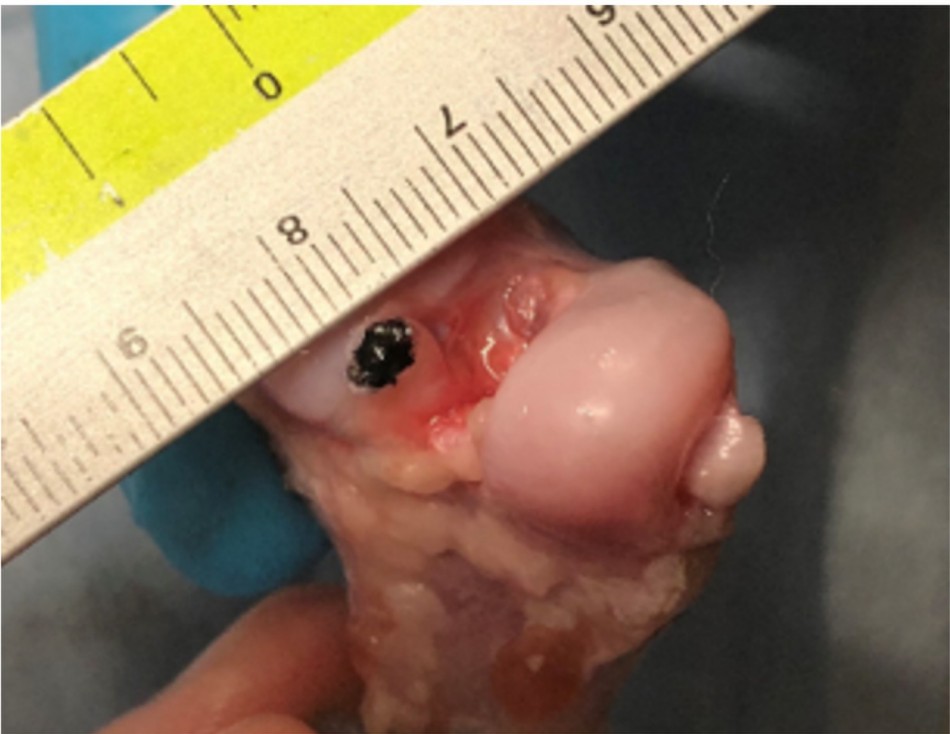

**Fig 3. Photograph used to measure the area of the femoral side in Image J.** The cranial cruciate ligament was transected perpendicular to the longitudinal axis of ligament with a scalpel at its attachments to the femur and tibia, and the section was stained with India ink to measure the cross-sectional area using Image J. The photograph was taken while the scale was taken to calibrate the object's length.

## Results

### Histological and immunohistological analysis

There was no macroscopic abnormality in the CrCL in either group at the time of euthanasia. The interfascicular area and fibrocartilage were more extensive at the ligament attachments, particularly on the tibial side than in the middle within each group (Fig 4). In addition, irregularities in ligamentous fibril orientation were observed on the tibial side. There were no significant differences in the cell density between the areas within each group and between the groups.

Areas with positive AB staining were particularly obvious in the areas of irregular fibril orientation around the round cells, such as in the interfascicular area and fibrocartilage (Fig 5). Within each group, the positive AB staining area ratio was significantly higher on the tibial side than on the femoral side and in the middle. The ratio was significantly higher in the control group than in the GC group on the tibial side.

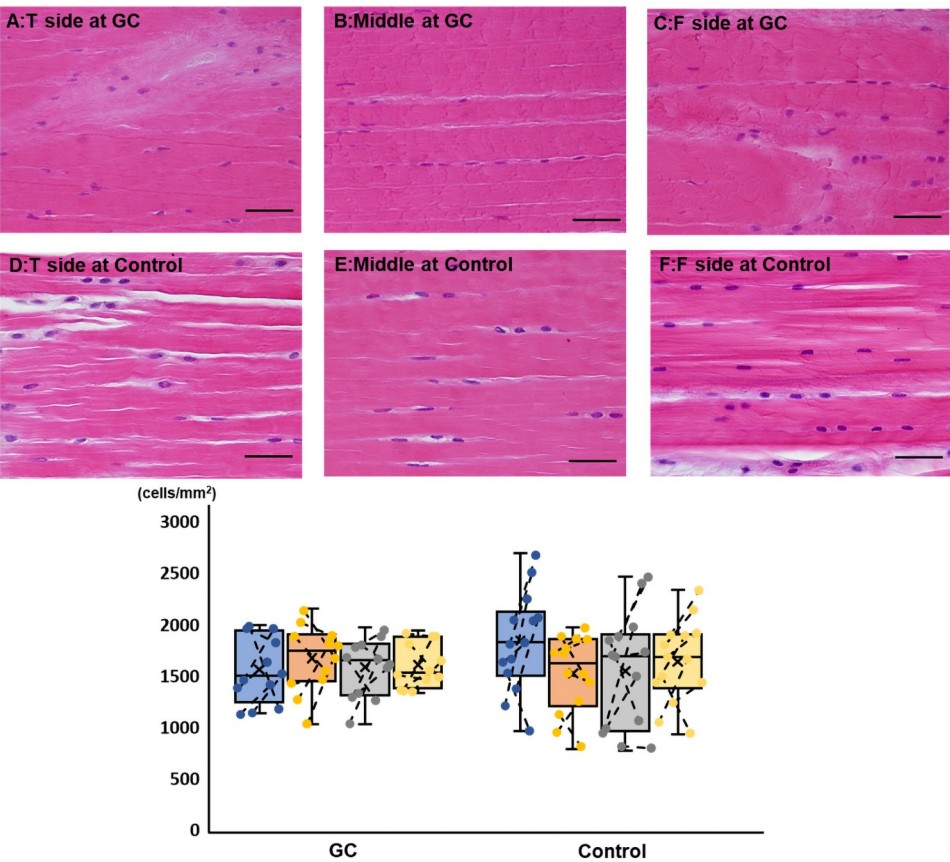

**Fig 4. Hematoxylin-eosin staining findings of the cranial cruciate ligament in the GC and control groups (bar = 20 μm).** The upper panels represent examples of staining on the tibial (A), middle (B), and femoral (C) sides of the GC group. The lower panels represent examples of staining on the tibial (D), middle (E), and femoral (F) sides of the control group. In the box-and-whisker plot, blue, orange, gray, and yellow indicate the tibial side, middle, femoral side, and the entire cell count, respectively. If the plots are connected by a line, they are stifle joints of the same dog. Within each group, the interfascicular area and fibrocartilage are more extensive at the ligament attachments, particularly on the tibial side, than in the middle. Irregularities are observed in fibril orientation. In the analysis of cell density, there are no significant differences between the areas within each group and between the groups. GC, glucocorticoid.

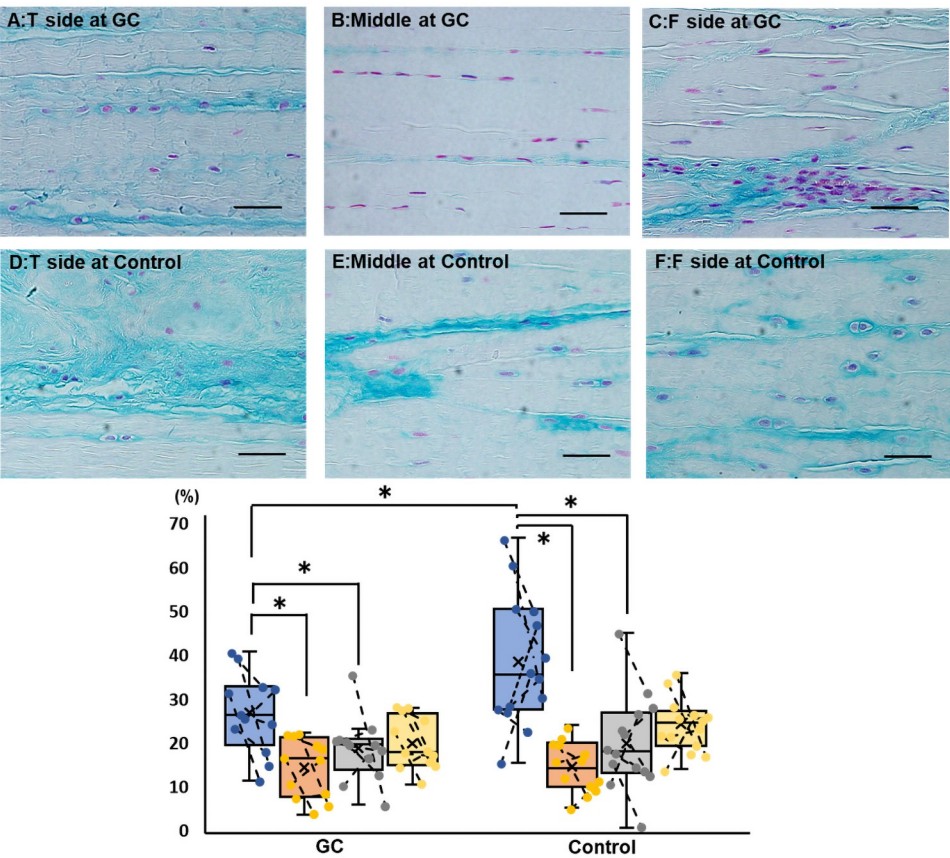

**Fig 5. Alcian-blue staining findings of the cranial cruciate ligament in the GC and control groups (bar = 20 μm).**
The upper panels represent examples of staining on the tibial (A), middle (B), and femoral (C) sides of the GC group.
The lower panels represent examples of staining on the tibial (D), middle (E), and femoral (F) sides of the control
group. In the box-and-whisker plot, blue, orange, gray, and yellow indicate the tibial side, middle, femoral side, and the
entire measurement, respectively. The asterisks indicate P<0.05. If the plots are connected by a line, they are stifle
joints of the same dog. Areas with positive Alcian-blue staining are particularly obvious in the areas of irregular fibril
orientation around the round cells, such as the interfascicular area and fibrocartilage. Within each group, the positive
Alcian-blue area ratio is significantly higher on the tibial side than on the femoral side and in the middle. The ratio is
significantly higher in the control group than in the GC group on the tibial side. GC, glucocorticoid.

Areas with positive EE staining were observed, particularly in the interfascicular area and
fibrocartilage (Fig 6). The positive EE area ratio was significantly higher on the tibial side than
on the femoral side and in the middle for the control group. The ratio was significantly higher
at the tibial, middle, femoral, and entire areas in the control group than in the GC group.

Extensive positive areas of COL1 immunohistochemical staining were found in areas with
normal ligament fibril orientation (Fig 7). While analyzing the positive ratio, there were no sig-
nificant differences between the areas within each group and between the groups.

The area ratio of positive elastin immunohistochemical staining was significantly higher on
the tibial side than on the femoral and the middle in the control group (Fig 8). The ratio was
significantly higher in the control group than in the GC group on the tibial side.

## Biomechanical analysis

The failure pattern in both groups demonstrated an avulsion fracture at the tibial attachment
in all specimens (Fig 9). One specimen from the control group had an avulsion fracture of the

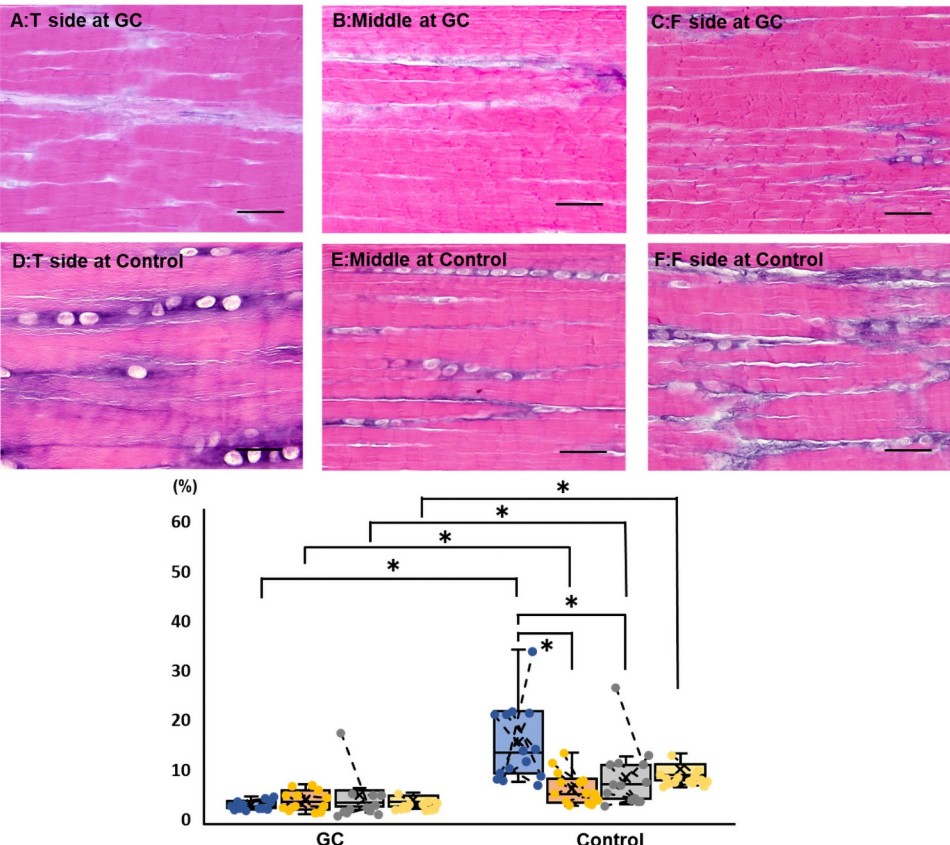

**Fig 6. Elastica-Eosin staining findings of the cranial cruciate ligament in the GC and control groups (bar = 20 µm).** The upper panels represent examples of staining on the tibial (A), middle (B), and femoral (C) sides of the GC group. The lower panels represent examples of staining on the tibial (D), middle (E), and femoral (F) sides of the control group. In the box-and-whisker plot, blue, orange, gray, and yellow indicate the tibial side, middle, femoral side, and the entire measurement, respectively. The asterisks indicate P<0.05. If the plots are connected by a line, they are stifle joints of the same dog. Positive areas of Elastica-Eosin are observed, particularly in the interfascicular area and fibrocartilage. In the control group, the ratio is significantly higher on the tibial side than on the femoral and middle. The ratio is significantly higher in the control group compared to the GC group in the tibial, middle, femoral, and entire areas. GC, glucocorticoid.

tibia with a partial ligament tear. The maximum tensile strength, maximum stress, elastic modulus, and strain at the transition point did not differ significantly between the groups. The stress at the transition point was significantly lower in the GC group than in the control group (Fig 10).

## Discussion

No previous study has examined the effects of high-dose administration of GC on the CrCL in dogs. The present study results indicate that a high-dose of prednisolone may affect metabolism, such as the production of mucopolysaccharides and elastic fibers, although the effect on COL1 production, the major ECM, is small. The change of ECM had a small effect on the strength of the ligament, but it was shown to affect the toe region. In conclusion, the ligamentous changes associated with GC were different from the typical degeneration observed in spontaneous CCLD.

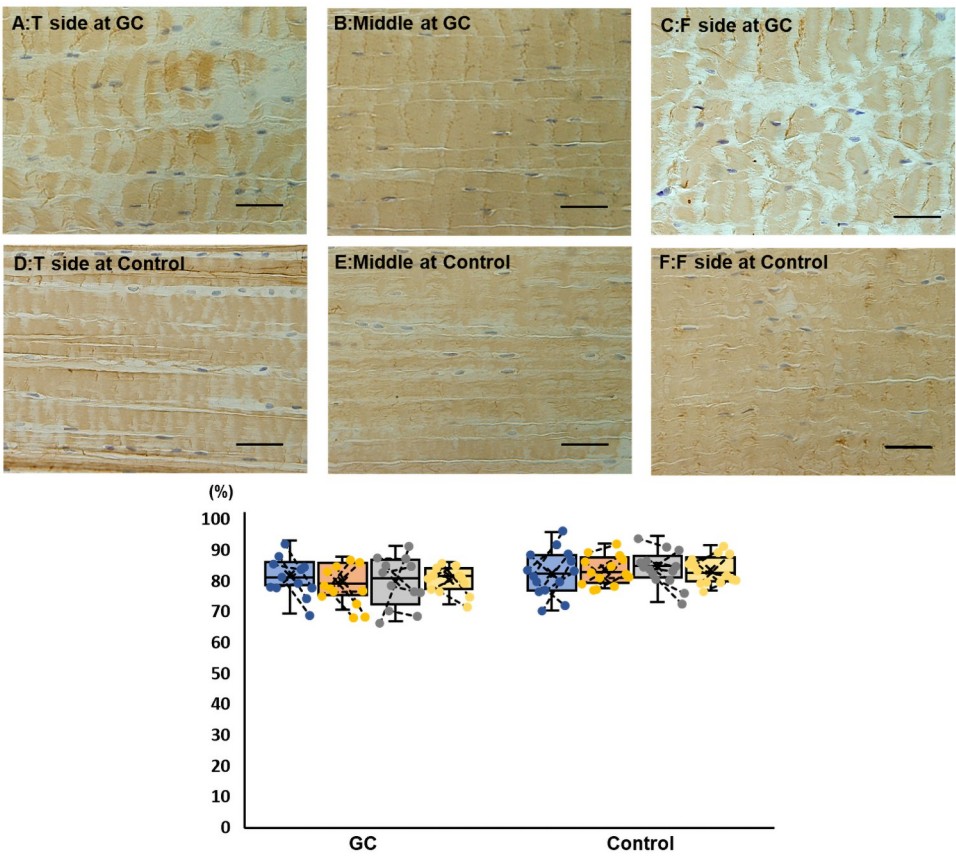

**Fig 7. Immunohistochemical staining of type 1 collagen findings of the cranial cruciate ligament in the GC and control groups (bar = 20 μm).** The upper panels depict examples of staining on the tibial (A), middle (B), and femoral (C) sides of the GC group. The lower panels depict examples of staining on the tibial (D), middle (E), and femoral (F) sides of the control group. In the box-and-whisker plot, blue, orange, gray, and yellow indicate the tibial side, middle, femoral side, and the entire measurement, respectively. If the plots are connected by a line, they are stifle joints of the same dog. Extensive positive regions are found in areas with normal ligament orientation. There are no significant differences between the areas within each group and between the groups in the positive ratio analysis. GC, glucocorticoid.

CCLD is characterized by a decrease in COL1 along with chondrometaplasia, in addition to an increase in mucopolysaccharides, a representative of the cartilage matrix [4–6]. In addition, recent reports have shown that elastic fibers increase with ligament degeneration in Labrador Retrievers [23]. Elastic fibers are considered a minor ECM in ligaments [7]. However, Smith et al. reported that they account for 9.86±3.97% of the dry weight in greyhound CrCLs [24]. Interestingly, greyhounds, a breed known to have a low incidence of CrCLR, have been reported to have more elastic fibers in their CrCL than those in Labrador Retrievers [23, 24]. Therefore, greyhounds may compensate for CrCL damage by producing elastic fibers. Elastic fibers may be associated with CrCL repair, and the difference in the balance between repair and damage may be reflected in the risk of CrCLR [23]. Despite the unclear pathogenesis of CCLD, researchers have identified numerous associated factors. In humans, non-traumatic Achilles tendon rupture has been reported to be associated with Cushing's syndrome and long-term steroid therapy [14, 15]. Similarly, hypercorticoidemia in horses is a risk factor for injury [25, 26]. These studies reported collagen misalignment and mucopolysaccharide

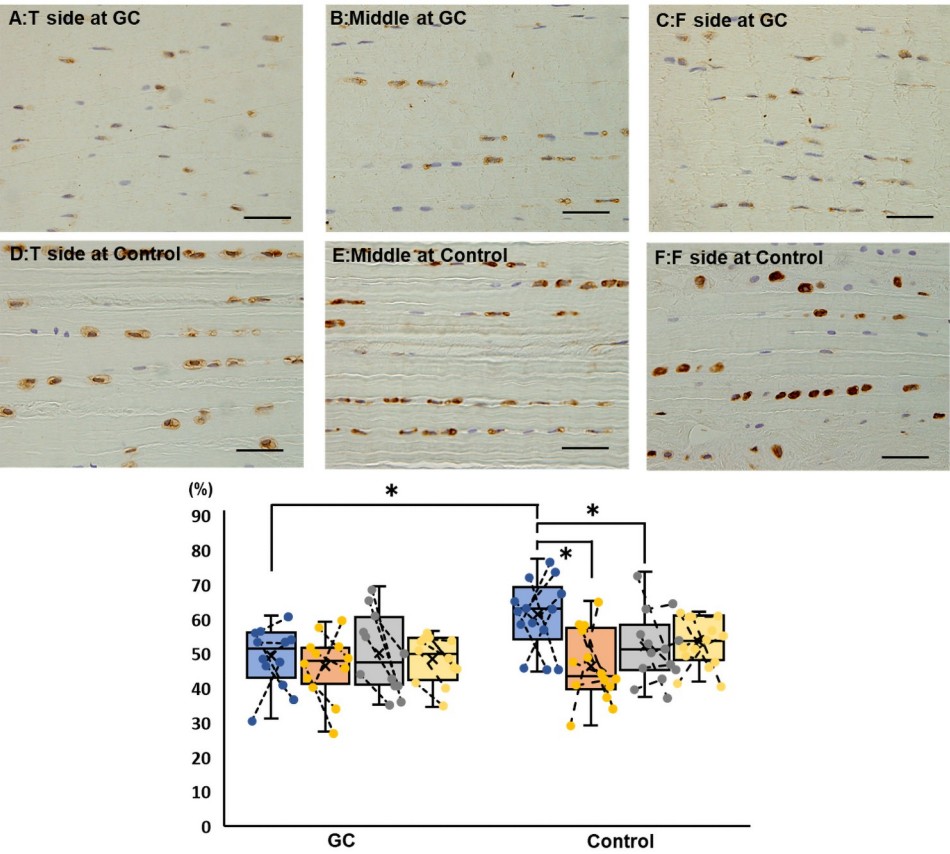

**Fig 8. Immunohistochemical staining of the elastin findings of the cranial cruciate ligament in the GC and control groups (bar = 20 μm).** The upper panels depict examples of staining on the tibial (A), middle (B), and femoral (C) sides of the GC group. The lower panels depict examples of staining on the tibial (D), middle (E), and femoral (F) sides of the control group. In the box-and-whisker plot, blue, orange, gray, and yellow indicate the tibial side, middle, femoral side, and the entire measurement, respectively. The asterisks indicate P<0.05. If the plots are connected by a line, they are stifle joints of the same dog. In the control group, the ratio is significantly higher on the tibial side than on the femoral and middle. The ratio is significantly higher in the control group than in the GC group on the tibial side. GC, glucocorticoid.

deposition, similar to canine CCLD, as features of suspensory ligament degeneration in horses with pituitary pars intermedia dysfunction [25, 26].

In both the GC and control groups, the increased production of mucopolysaccharides was observed in areas where there were irregularities in ligament orientation in the ligamentous attachments of the tibia. There were more elastin-positive cells and elastic fibers in the tibial attachments than in other regions in the control group. More elastic fibers were produced in areas with pronounced irregular fibril orientation in the control group, such as tibial attachments. In contrast, fewer elastic fibers were produced throughout the ligament in the GC group compared to the control group. In this study, CCLD-like degeneration was expected to occur as in the equine suspensory ligament, but no deposition of mucopolysaccharides associated with hypercorticoidemia was observed. In vitro studies have reported that the administration of GC preparations to tenocytes inhibits the production of collagen and proteoglycans [27, 28]. The production of ECM components was possibly suppressed in this study, and these effects, but the effect on COL1 was minimal. The glucocorticoid receptor is down-regulated with long-term GC administration in dogs and humans [29, 30]. However, the number of

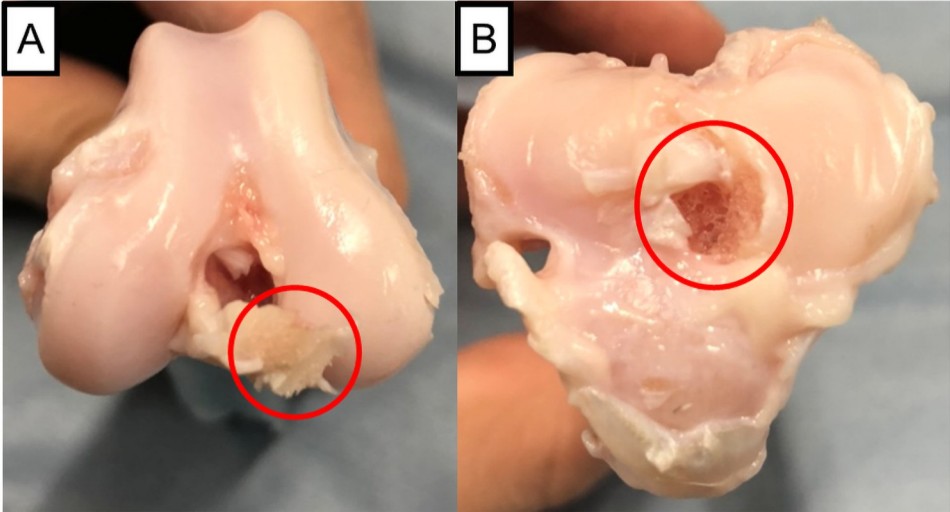

**Fig 9. Failure pattern of the biomechanical test.** A: The femoral side after the failure. The end of the ligament is attached to the bony fragment of the tibia (red circle). B: The tibial side after the failure. The cranial cruciate ligament is detached over the entire area (red circle).

glucocorticoid receptor-positive cells rather increased in the suspensory ligament of horses with pituitary pars intermedia dysfunction [26]. Differences in the changes associated with hypercortisolemia in horses may be attributed to the differences in exposure to blood GC concentrations between the endocrine disease with chronic hypercorticoidemia and the experimental model with high-dose GC preparations and the exercise restriction-induced limitation of chronic mechanical loading on the ligaments. This might have resulted in a different mechanism for CCLD-like degeneration. Thus, various factors are intricately involved in the

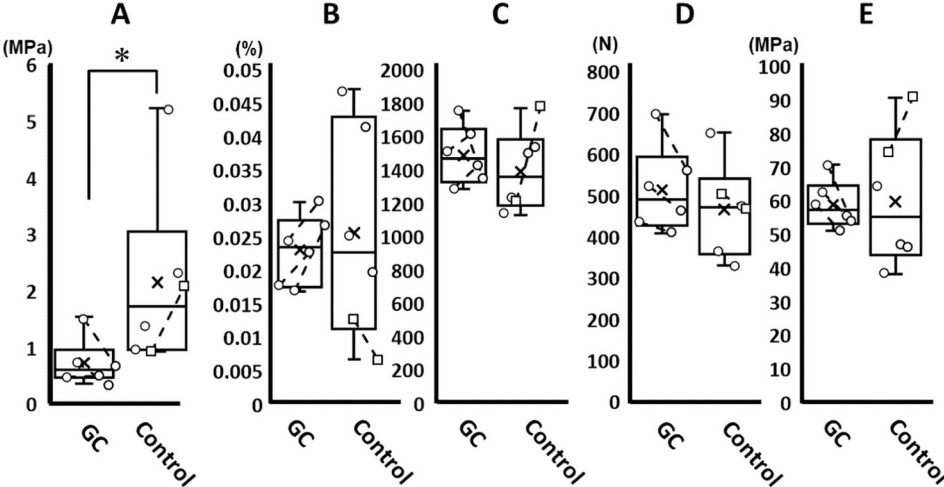

**Fig 10. Tensile test results of the bone-ligament-bone model in the GC and control groups.** A: stress at the transition point; B: strain at the transition point; C: elastic modulus; D: maximum tensile strength; and E: maximum stress. The asterisks indicate P<0.05. If the plots are connected by a line, they are stifle joints of the same dog. The stress at the transition point is significantly lower in the GC group than in the control group. GC, glucocorticoid.

bioactivity of GCs and their regulation. Moreover, it is still unclear if the abovementioned events occur in all tissues similarly.

The failure of all specimens in the biomechanical analysis is attributed to the detachment at the tibial CrCL attachment. A previous Rottweilers and Greyhounds report showed that 8 out of 14 stifles were detached at the tibial CrCL attachment after the axial loading test [31]. This may be a consequence of changes in the ECM component at the tibial attachment. The transition occurs between ligament and cartilage with changes in material properties, thus causing stress concentration [32, 33]. Therefore, in young, healthy beagle dogs with minimal ligament weakening associated with CCLD, stress was thought to be concentrated at the ligament origin, and detachment occurred. A significant difference in stress at the transition point was observed, similar to the results reported using the porcine medial collateral ligament [34]. In other words, the transition between the toe region and liner region was smooth, owing to the presence of elastic fibers in the control group. Considering the absence of elastic fibers in the GC group, it was impossible to provide sufficient pre-stress to the collagen fibers, suggesting the risk of micro-damage to the ligament. Thus, our results suggest that CrCL changes associated with high-dose GC administration may be different from that of canine CCLD, which in turn is characterized by the chondrogenesis of the CrCL.

Our study had several limitations. First, different beagle dogs were used for histological and biomechanical studies. The relationship between the histological and biomechanical analysis could have been directly examined using one side for each analysis; however, beagle dogs created for other purposes were used in this study. As a background to this study design, we conducted a histological study while GC was unclear about its effect on CrCL. As a result, we could recognize the changes associated with GC. The year after we completed this study, we learned that we would be creating three of these model dogs again. We planned to investigate how the histological effects would affect the biomechanical effects. This study collected ligaments in the control group from dogs euthanized for non-orthopedic studies and dogs used on one side for surgical training in veterinary students. For the histological and biomechanical studies, specimens collected at close time points were used as controls. Only one side was available in the biomechanical study as four of the five dogs were used for surgical training. Thus, we also handled the stifle joints from the same dog as independent data. Second, it was difficult to set up the ligament on the universal material testing machine for the biomechanical study. Thus, the femur and tibia needed to be clamped for the experiment. Moreover, it was impossible to thoroughly investigate the material properties of the ligament because of the bone detachment that occurred at the CrCL attachment. In addition, The specimens were frozen once and then thawed for the experiment. Since previous reports have shown that freezing and thawing have little effect on the biomechanical characteristics of the patellar tendon [35], the effects of freezing and thawing in the results are expected to be small. Third, the cross-sectional area of the ligament was calculated after the experiment. Thus, it may differ from the original cross-sectional area because of elongation. Fourth, higher doses and longer durations of GCs were administered than those used in clinical practice to clarify the effects of GCs. Therefore, our results do not necessarily reflect Cushing's syndrome in clinical cases or the degeneration of the CrCL in the case of long-term GC administration.

Our findings suggested that high-dose GC treatment for 84 days in healthy dogs suppressed the production of mucopolysaccharides and elastic fibers, although the effect on COL1 production was small. The change of ECM had a small effect on the strength of the ligament, but it was shown to affect the toe region. In conclusion, the ligamentous changes associated with GC were different from the degeneration observed in spontaneous CCLD. Even though the role of elastic fibers is still uncertain, it is unclear to what extent prolonged exposure to

hypercorticoidemia is a risk factor for CrCLR in dogs. Future epidemiological studies are expected to clarify the relationship between CrCLR and hypercorticoidemia.

## Supporting information

**S1 File. Raw data.**
(XLSX)

## Acknowledgments

We gratefully acknowledge the work of the past and present members of our laboratory. We also thank Editage (www.editage.com) for their English language editing service.

## Author Contributions

**Conceptualization:** Yasushi Hara.

**Data curation:** Masakazu Shimada, Koki Wada, Sawako Murakami.

**Formal analysis:** Masakazu Shimada, Koki Wada, Sawako Murakami, Nobuo Kanno.

**Funding acquisition:** Yasushi Hara.

**Investigation:** Masakazu Shimada, Koki Wada, Sachiyo Tanaka, Sawako Murakami, Nobuo Kanno.

**Methodology:** Masakazu Shimada, Sachiyo Tanaka, Nobuo Kanno, Yasushi Hara.

**Project administration:** Masakazu Shimada.

**Resources:** Masakazu Shimada, Nobuo Kanno, Yasushi Hara.

**Supervision:** Kei Hayashi, Yasushi Hara.

**Validation:** Nobuo Kanno, Kei Hayashi, Yasushi Hara.

**Visualization:** Masakazu Shimada, Koki Wada.

**Writing – original draft:** Masakazu Shimada.

**Writing – review & editing:** Masakazu Shimada, Sawako Murakami, Nobuo Kanno, Kei Hayashi, Yasushi Hara.

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
