## [Decision Letter · Decision Letter 0]

19 Oct 2021

PONE-D-21-25951Effects of long-term and high-dose administration of glucocorticoids on the cranial cruciate ligament in healthy beagle dogsPLOS ONE

Dear Dr. Shimada,

Thank you for submitting your manuscript to PLOS ONE. After careful consideration, we feel that it has merit but does not fully meet PLOS ONE’s publication criteria as it currently stands. Therefore, we invite you to submit a revised version of the manuscript that addresses the points raised during the review process.

Many thanks for submitting your manuscript to PLOS One

It was reviewed by two experts in the field, and they have recommended some modifications be made prior to acceptance

I therefore invite you to make these changes and to write a response to reviewers which will expedite revision upon resubmission

I wish you the best of luck with your modifications

Hope you are keeping safe and well in these difficult times

Thanks

Simon

We look forward to receiving your revised manuscript.

Kind regards,

Simon Clegg, PhD

Academic Editor

PLOS ONE

Journal Requirements:

2. In your Methods section, please provide additional details regarding the animals used in your study and ensure you have described the source. For more information regarding PLOS' policy on materials sharing and reporting, see https://journals.plos.org/plosone/s/materials-and-software-sharing#loc-sharing-materials.

Reviewers' comments:

Reviewer's Responses to Questions

**Comments to the Author**

1. Is the manuscript technically sound, and do the data support the conclusions?

Reviewer #1: Partly

Reviewer #2: No

Reviewer #3: Partly

2. Has the statistical analysis been performed appropriately and rigorously? 

Reviewer #1: No

Reviewer #2: Yes

Reviewer #3: Yes

3. Have the authors made all data underlying the findings in their manuscript fully available?

Reviewer #1: No

Reviewer #2: No

Reviewer #3: Yes

4. Is the manuscript presented in an intelligible fashion and written in standard English?

Reviewer #1: Yes

Reviewer #2: Yes

Reviewer #3: Yes

5. Review Comments to the Author

Reviewer #1: The manuscript describes the results from a highly relevant study – the investigation of the influence of high-dose glucocorticoids (GC) on ligament composition and biomechanics in dogs with the cranial cruciate ligament as a model. The results are clinically highly relevant.

There are two major concerns. First, there is an ethical aspect of the study. Although the study was approved by local Bioethics Committee of the Experimental Animal Committee, the dogs were held for almost 3 months in small cages of 0.84 m2 with no other exercise allowed outside the cages.

Two, handling of the statistics. In the biomechanical analysis 6 stifles in the GC treated group and 6 stifles in the control group. However, the 6 stifles originated from only 3 dogs of the GC-treated dogs and from 5 different dogs in the control group. In the statistical analysis, the biomechanical measurements are treated as independent data. This must be addressed, or at least commented in the text.

Row 40-41: Include a reference for the statement that cranial cruciate ligament rupture is a common (stifle) disease in dogs.

Row 41-42: “Canine CrCLR is characterized by progressive chronic ligament degeneration with chondrometaplasia, and is referred to as…” The referred articles only support the histological characteristics of already ruptured cranial cruciate ligament, and that it can be expected to have extensive changes in a ligament after rupture. Additional references are needed to support both the presence of a progressive chronic ligament degeneration before rupture (as in ref 4. Kyllar et al 2018), and that it is predisposing for rupture) or rewrite the sentence in a way that differentiate between what is known and what is assumed.

Row 52-54 When the authors refer to glucocorticoid effects in the body it is important to differentiate between physiological effects and the effects from supraphysiologic doses of glucocorticoids. Rewrite.

Row 58 When the authors state “An association between hypercortisolemia such as hypercortisolism, and CrCLR in dogs has been reported”, the authors refer to an old edition of a book on veterinary endocrinology. In that book, it is said that “Chronic hypercortisolism can result in an exaggeration of common problems such as anterior cruciate ligament rupture and patellar luxation lameness”. Rewrite the sentence and/or find another more adequate reference.

Row 74-75 There are 12 control dogs but only 20 cranial cruciate ligaments. What happened to the missing 4?

Row 77 Why were only 6 stifles from 5 of the control dogs used?

Table 1 In this group of small number of dogs, it is most appropriate to present body weight and age as median and range.

Rows 121-129 Specimen preparation Is this a standard procedure to test ligament biomechanics? If so – please give appropriate reference(s). If not, please include information about possible influences of freezing and thawing on the test result.

Rows 131-146 Testing protocol. Is this a standard protocol to test the biomechanics of the cranial cruciate ligament in dogs? If so – please give appropriate reference(s). If not, please include information about how the choices of joint angle and procedure were made.

Rows 161-165: Statistical testing. How has the authors handled the fact that each dog contributes with 2 cranial cruciate ligaments for histological analysis and that the 6 stifles originate from only 3 dogs?

Figures 3-6, 8 and 10. Considering the small number of dogs, replace the box-plots with Jitter-plots. This way it would also be possible to indicate which pair of measure points that originates from the same dog. Statistical testing might need correction.

Reviewer #2: The work is interesting and reveals the use of a biological model to understand relevant metabolic changes. It uses techniques that reveal the modulation and structuring of joint ligaments. Above all, it makes analogies with data from the literature that could not be compared with each other, as explained in lines 287-302. Because they are different structures in their architecture and function. Otherwise, when framing the work properly, comparing ligaments with each other, and not ligaments with muscles or tendons, the work can be re-evaluated. Biomechanical data must be compared with relevant literature.

Reviewer #3: The aim of this study was to determine the effects of long-term, high-dose glucocorticoids (GCs) administration on the histological and mechanical properties of the cranial cruciate ligament (CrCL) in healthy Beagle dogs. The study looks interesting and was able to prove that the changes caused by longterm use on the cranial cruciate ligament differ from those seen in cranial cruciate ligament disease.

Even so, there were limitations in the study that prevented these effects from being concluded. Some doubts were observed in the study and I would like the authors to justify:

1. What is the study or criteria adopted to consider the use of GC for three months as long term and 2mg/kg 12/12 hours as high doses?

2. Does the methodology state that the corticoid was used for 84 days, but in the last paragraph of the discussion, did the authors mention three months? How to explain?

3. I consider the form of euthanasia of dogs questionable with only an overdose of pentobarbital and without the associated use of a cardioplegic such as potassium chloride.

4. Although the CG and control groups remained in a cage, without any other type of exercise (standardization), was the time these dogs remained in a cage not mentioned? Could restricted movement harm a healthy joint and its compound ligaments? How would the authors justify this?

5. Is there any study on the long-term use of GC and the incidence of cranial cruciate ligament rupture in a healthy joint that would justify the study?

6. PLOS authors have the option to publish the peer review history of their article (what does this mean?). If published, this will include your full peer review and any attached files.

Reviewer #1: No

Reviewer #2: No

---

## [Author Response · Author response to Decision Letter 0]

2 Dec 2021

Reviewer #1: The manuscript describes the results from a highly relevant study – the investigation of the influence of high-dose glucocorticoids (GC) on ligament composition and biomechanics in dogs with the cranial cruciate ligament as a model. The results are clinically highly relevant.

 Response:　 Thank you for your comments, which were highly insightful and enabled us to improve the quality of our manuscript significantly.

There are two major concerns. First, there is an ethical aspect of the study. Although the study was approved by local Bioethics Committee of the Experimental Animal Committee, the dogs were held for almost 3 months in small cages of 0.84 m2 with no other exercise allowed outside the cages.

 Response:　 As for the activities, we misleadingly wrote some of them. We kept them basically in the cage, but we let them go indoors for about an hour twice daily (morning and the evening). However, the dogs used in this study were not given any other exercise load, such as a treadmill. The GC group of dogs used in this study was created for cardiovascular research, while the control group was used for unrelated research and surgical training of veterinary students. Therefore, no exercise loading conditions were set. (Lines 90-92)

Two, handling of the statistics. In the biomechanical analysis 6 stifles in the GC treated group and 6 stifles in the control group. However, the 6 stifles originated from only 3 dogs of the GC-treated dogs and from 5 different dogs in the control group. In the statistical analysis, the biomechanical measurements are treated as independent data. This must be addressed, or at least commented in the text.

 Response:　 We wrote about the points you pointed out in the limitation section. (Lines 366-376)

Row 40-41: Include a reference for the statement that cranial cruciate ligament rupture is a common (stifle) disease in dogs.

 Response:　We have included two references for the statement. (Line 41)

Row 41-42: “Canine CrCLR is characterized by progressive chronic ligament degeneration with chondrometaplasia, and is referred to as…” The referred articles only support the histological characteristics of already ruptured cranial cruciate ligament, and that it can be expected to have extensive changes in a ligament after rupture. Additional references are needed to support both the presence of a progressive chronic ligament degeneration before rupture (as in ref 4. Kyllar et al 2018), and that it is predisposing for rupture) or rewrite the sentence in a way that differentiate between what is known and what is assumed.

 Response:　 We have added two references. (Line 43)

Row 52-54 When the authors refer to glucocorticoid effects in the body it is important to differentiate between physiological effects and the effects from supraphysiologic doses of glucocorticoids. Rewrite.

 Response:　 We rewrote it as “The physiological effects of GCs include metabolism, water and electrolyte balance, and immunomodulation[12, 13]. The effects from supraphysiologic doses of glucocorticoids include anti-inflammatory, immunosuppressive, and protein catabolic effects [12, 13].” (Lines 53-55)

Row 58 When the authors state “An association between hypercortisolemia such as hypercortisolism, and CrCLR in dogs has been reported”, the authors refer to an old edition of a book on veterinary endocrinology. In that book, it is said that “Chronic hypercortisolism can result in an exaggeration of common problems such as anterior cruciate ligament rupture and patellar luxation lameness”. Rewrite the sentence and/or find another more adequate reference.

 Response:　We rewrote “The endocrinology textbook said chronic hypercortisolism could result exaggerate common problems such as CrCLR and patellar luxation lameness [17].”　 (Lines 59-61)

Row 74-75 There are 12 control dogs but only 20 cranial cruciate ligaments. What happened to the missing 4?

Row 77 Why were only 6 stifles from 5 of the control dogs used?

 Response:　This study collected ligaments from model dogs euthanized for cardiovascular studies, dogs euthanized for non-orthopedic studies, and dogs used on one side for surgical training in veterinary students. For the histological and biomechanical studies, specimens collected at close time points were used as controls. Only one side was available in the biomechanical study as four of the five dogs were used for surgical training. (Lines 76-77, 366-376)

Table 1 In this group of small number of dogs, it is most appropriate to present body weight and age as median and range.

Response:　We have corrected body weight and age to the median and range. (Table 1)

Rows 121-129 Specimen preparation Is this a standard procedure to test ligament biomechanics? If so – please give appropriate reference(s). If not, please include information about possible influences of freezing and thawing on the test result.

 Response:　 We discussed this in the limitation section and cited a paper that showed that the effect of freezing and thawing is minimal. (Lines 380-383)

Rows 131-146 Testing protocol. Is this a standard protocol to test the biomechanics of the cranial cruciate ligament in dogs? If so – please give appropriate reference(s). If not, please include information about how the choices of joint angle and procedure were made.

 Response:　 The joint was set to 135⁰, the angle in the standing position, and the CrCL was placed on the machine, the same direction as the proximal-distal direction of distraction. The creep release is incorrectly stated as 5N, not 0.5N. My apologies. This was according to previous reports. (Lines 146-148)

Rows 161-165: Statistical testing. How has the authors handled the fact that each dog contributes with 2 cranial cruciate ligaments for histological analysis and that the 6 stifles originate from only 3 dogs?

 Response:　 In this study, paired stifles samples were not necessarily available; therefore, they were used as independent data. As a background to this study design, we conducted a histological study while GC was unclear about its effect on CrCL. Thus, we could recognize the changes associated with GC. The year after we completed this study, we learned that we would be creating three of these model dogs again. We planned to investigate how the histological effects would affect the biomechanical effects. As mentioned in the Limitation, it would be ideal to use one side as the tissue and one side as the biomechanical experiment in this study, which is the reason for the design of this study. (Lines 366-376)

Figures 3-6, 8 and 10. Considering the small number of dogs, replace the box-plots with Jitter-plots. This way it would also be possible to indicate which pair of measure points that originates from the same dog. Statistical testing might need correction.

 Response:　 All figures have been corrected. In this study, paired stifles samples were not necessarily available; therefore, they were used as independent data.

Reviewer #2: The work is interesting and reveals the use of a biological model to understand relevant metabolic changes. It uses techniques that reveal the modulation and structuring of joint ligaments. Above all, it makes analogies with data from the literature that could not be compared with each other, as explained in lines 287-302. Because they are different structures in their architecture and function. Otherwise, when framing the work properly, comparing ligaments with each other, and not ligaments with muscles or tendons, the work can be re-evaluated. Biomechanical data must be compared with relevant literature.

Response: Thanks for the comments, which were highly insightful and enabled us to improve our manuscript's quality significantly. To the best of my knowledge, there are no confirmed epidemiological or experimental studies on the relationship between GC and CCLD in dogs. Therefore, we compared the histological study with previous CCLD studies and considered suspensory ligament degeneration of horses with pituitary pars intermedia dysfunction. In the biomechanical study, to discuss the histologically revealed changes of Elastin in CrCL, we compared the results with the study of biomechanical effects of Elastin in the porcine medial collateral ligament.

Reviewer #3: The aim of this study was to determine the effects of long-term, high-dose glucocorticoids (GCs) administration on the histological and mechanical properties of the cranial cruciate ligament (CrCL) in healthy Beagle dogs. The study looks interesting and was able to prove that the changes caused by longterm use on the cranial cruciate ligament differ from those seen in cranial cruciate ligament disease.

Even so, there were limitations in the study that prevented these effects from being concluded. Some doubts were observed in the study and I would like the authors to justify:

Response: Thanks for your comments, which were highly insightful and enabled us to improve our manuscript’s quality significantly.

1. What is the study or criteria adopted to consider the use of GC for three months as long term and 2mg/kg 12/12 hours as high doses?

 Response: This study used ligaments from a model created for cardiovascular research. A previous report examined a model in which dogs were administered a high dose of a synthetic corticosteroid at 2 mg/kg every 12 h for 28 days, which showed echocardiographic cardiac morphology and function changes, but did not reveal any abnormalities histological changes. Past studies have suggested that chronic hypercortisolism can affect cardiac function and morphology. Our research group hypothesized that histological changes would appear with longer-term observation and conducted the study for 84 days. Thus, echogenicity and histological changes such as fibrosis of the myocardium were observed. (Lines 76-82)

2. Does the methodology state that the corticoid was used for 84 days, but in the last paragraph of the discussion, did the authors mention three months? How to explain?

 Response:　 In previous reports, the study was conducted for 28 days, set as 1month for convenience. Therefore, 84 days was described as 3 months. To avoid confusion, we have changed 3 months to 84 days. (Lines 21, 389)

3. I consider the form of euthanasia of dogs questionable with only an overdose of pentobarbital and without the associated use of a cardioplegic such as potassium chloride.

Response:　 The method of euthanasia was selected based on the AVMA guidelines. For euthanasia, all dogs were confirmed to have cardiac arrest by ECG monitoring and auscultation at least 5 minutes and respiratory arrest for at least 5 minutes. Then, the corneal reflex was performed to certify death. 

4. Although the CG and control groups remained in a cage, without any other type of exercise (standardization), was the time these dogs remained in a cage not mentioned? Could restricted movement harm a healthy joint and its compound ligaments? How would the authors justify this?

Response:　 We did not restrict exercise to protect the ligament. This study collected ligaments from model dogs euthanized for cardiovascular studies, dogs euthanized for non-orthopedic studies, and dogs used on one side for surgical training. We kept them basically in the cage, but we let them go indoors for about an hour twice daily (morning and the evening). However, the dogs used in this study were not given any other exercise load, such as a treadmill. Therefore, we consider that the exercise load was almost the same for all dogs.

5. Is there any study on the long-term use of GC and the incidence of cranial cruciate ligament rupture in a healthy joint that would justify the study?

Response: As mentioned in the introduction and discussion, no studies have associated CrCLR with long-term administration. However, considering the supraphysiological effects of GCs, there is a reasonable risk that they may be affected. Due to the need to minimize the creation of model animals due to ethical aspects, we experimented because we thought it would be an opportunity to create such a model in other studies and clarify the effects of GC.

---

## [Editor Report · Decision Letter 1]

20 Dec 2021

Effects of long-term and high-dose administration of glucocorticoids on the cranial cruciate ligament in healthy beagle dogs

PONE-D-21-25951R1

Dear Dr. Shimada,

We’re pleased to inform you that your manuscript has been judged scientifically suitable for publication and will be formally accepted for publication once it meets all outstanding technical requirements.

Kind regards,

Simon Clegg, PhD

Academic Editor

PLOS ONE

Additional Editor Comments:

Many thanks for resubmitting your manuscript to PLOS One

As you have addressed all the comments and the manuscript reads well, I have recommended it for publication

You should hear from the Editorial Office shortly.

It was a pleasure working with you and I wish you the best of luck for your future research

Hope you are keeping safe and well in these difficult times

Thanks

Simon

---

## [Editor Report · Acceptance letter]

24 Dec 2021

PONE-D-21-25951R1 

Effects of long-term and high-dose administration of glucocorticoids on the cranial cruciate ligament in healthy beagle dogs 

Dear Dr. Shimada:

I'm pleased to inform you that your manuscript has been deemed suitable for publication in PLOS ONE. Congratulations! Your manuscript is now with our production department. 

Kind regards, 

on behalf of

Dr. Simon Clegg 

Academic Editor

PLOS ONE